TOPICAL REVIEW

# The central benefit of physiologically induced ketogenic states

Antonia Giacco[1], Giuseppe Petito[2], Rosalba Senese[2] (iD), Maria Moreno[1], Assunta Lombardi[3],
Antonia Lanni[2] (iD) and Pieter de Lange[2] (iD)

[1]*Dipartimento di Scienze e Tecnologie, Università Degli Studi del Sannio, Via De Sanctis, Benevento, Italy*
[2]*Dipartimento di Scienze e Tecnologie Ambientali, Biologiche e Farmaceutiche, Università Degli Studi Della Campania 'Luigi Vanvitelli, Via Vivaldi, Caserta, Italy*
[3]*Dipartimento di Biologia, Università Degli Studi di Napoli 'Federico II,' Monte Sant'AngeloUniversità degli Studi di Napoli 'Federico II,' Monte Sant'Angelo, Naples, Italy*

Handling Editors: Karyn Hamilton & Val Andrew Fajardo

The peer review history is available in the Supporting Information section of this article (https://doi.org/10.1113/JP287462#support-information-section).

**Abstract figure legend** This review describes the association between ketogenic states induced by physiological interventions such as fasting or aerobic exercise, exogenous ketone supply and ketogenic diets, as well as increased cognition through increased activity of brain-derived neurotrophic factor (BDNF). The involvement of liver, intestine and muscle in this process is discussed, as well as the role of peroxisome proliferator-activated receptor (PPAR)δ in nutrient sensing and guidance of ketones to the brain.

**Abstract** Ketones not only act as metabolic fuel for the brain in periods of carbohydrate shortage, but also serve as signalling molecules that improve cognition. Ketogenic states can be induced peripherally by physiological interventions such as fasting and exercise, or ketogenic diets/exogenous supplementation. These interventions beneficially act on the brain through partially overlapping peripheral metabolic pathways. We focus on the role of peripheral organs such as the intestine, liver and skeletal muscle in mediating cognitive benefits in response to these interventions and discuss the prominent roles of the nuclear receptor peroxisome proliferator-activated receptor $\delta$, which serves as a nutrient sensor guiding ketones to the brain, where they stimulate the multifunctional cognition-improving factor, brain-derived neurotrophic factor.

(Received 14 October 2024; accepted after revision 3 April 2025; first published online 30 April 2025)

**Corresponding author** P. de Lange, Dipartimento di Scienze e Tecnologie Ambientali, Biologiche e Farmaceutiche, Università degli Studi della Campania 'Luigi Vanvitelli', Via Vivaldi, Caserta, Italy. Email: pieter.delange@unicampania.it

## Introduction

The induction of a ketogenic state has long been recognized to have beneficial peripheral and central outcomes. The early recognition that ketogenesis during the fasting state has anticonvulsant effects led to the application of ketogenic diets (KD) in the treatment of epilepsy in the 1920s (Wheless, 2008). More recently, ketones have been recognized to possess neuroprotective properties in a variety of diseases including Parkinson's disease, Huntington's disease, epilepsy and Alzheimer's disease (Maalouf et al., 2009), as well as during ageing (Acuña-Catalán et al., 2024). Ketogenesis is triggered by a metabolic switch that causes increased mobilization of fatty acids to serve as fuel over glucose (Anton et al., 2018). Importantly, the switch toward the use of ketones as energy molecules during periods of energy deprivation such as fasting or aerobic exercise implies cross-talk between interconnecting tissues including liver, the intestine, skeletal muscle and brain. Ketones are mainly produced by the liver, although an additional source for serum ketone levels in response to fasting is provided by the intestine (Bass et al., 2024). Increased adipose lipolysis leads to accumulation of adipose-derived fatty acids in the serum, which serve as crucial metabolites and signalling molecules. One important factor in the communication between energy deprivation, adipose fatty acid release, hepatic fatty acid oxidation and subsequent ketone transport into the brain is a member of the peroxisome proliferator-activated receptor (PPAR) nuclear receptor superfamily: PPAR$\delta$ (Chasseigneaux et al., 2024; Moreno et al., 2010). PPAR$\delta$ is expressed in skeletal muscle, liver, white adipose tissue and the CNS both in rodents and humans (Moreno et al., 2010). PPAR$\delta$ CNS expression extends to astrocytes, microglia, neurons and endothelial cells (Chasseigneaux et al., 2024; Schnegg & Robbins, 2011) from mouse to human (Warden et al., 2016). Fatty acids, the natural PPAR ligands, derive from ingested food (e.g. in the case of KDs) or are adipose-derived (e.g. in response to fasting through increased lipolysis) and directly control PPAR activity. Particularly, PPAR$\delta$ controls various pathways involved in metabolism in response to nutritional and physiological stimuli, including fatty acid oxidation (Moreno et al., 2010). Physiological interventions such as exercise or fasting activate PPAR$\delta$ by altering the supply of fatty acids to the tissues, including skeletal muscle muscle and brain (Moreno et al., 2010). In exercised muscle, ketones taken up from the serum attenuate muscle proteolysis (Evans et al., 2017) and fatty acid- activated PPAR$\delta$ increases the expression of target genes including those involved in mitochondrial dynamics (Chan et al., 2024). These

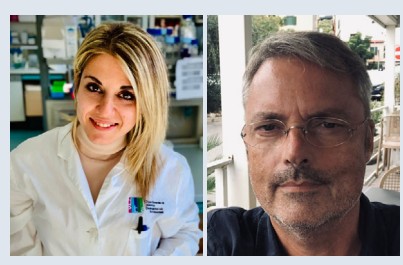

**Assistant Professor Antonia Giacco** obtained her PhD in Physiology from University of Sannio, Benevento, and **Pieter de Lange** is Associate Professor of Physiology at the University of Campania 'Luigi Vanvitelli,' Caserta. Both authors, as well as all of the coauthors, take part in a research group which also involves the University of Naples 'Federico II.' Over the past 30 years, all authors have focused on energy metabolism including mitochondrial bioenergetics and cellular signal transduction, and Antonia Giacco and Pieter de Lange, have studied the effect of fasting and exercise on metabolism in rodents. Recently, the group's research on peripheral organs has been extended to the brain, and a current research project involving Pieter de Lange and Assunta Lombardi continues to focus on the relationship between ketogenic states and the organism's peripheral and central metabolic health.

ketone- and fatty acid-related events are crucial for muscle integrity, by safeguarding the release of myokines, which is of importance in the prevention of reduced cognition during ageing (Oudbier et al., 2022). During prolonged fasting, termed Phase 2, ∼67% of the energy needed by the brain is provided by the ketone bodies beta hydroxy butyrate (BHB) and acetoacetate (Cahill, 2006). Prolonged voluntary exercise has been shown to increase circulating BHB in mice (Marosi et al., 2016; Sleiman et al., 2016). Serum BHB increases up to 2 mM after 2 days of fasting and after prolonged exercise, and up to 7 mM after 2–3 weeks of fasting, whereas after 3 weeks of a KD serum BHB levels can reach 5 mM (Kolb et al., 2021). Not only do ketones such as BHB serve as metabolic fuel, but also they induce pleiotropic metabolic changes in various organs by influencing substrate choice, inflammation and catabolism, as well as by modulating oxidative stress and gene expression (Newman and Verdin, 2014; Poff et al., 2020; Puchalska & Crawford, 2021). One important event related to the induction of ketosis is the activation of the class III histone deacetylase (HDAC) sirtuin1 (SIRT1) that boosts the shift toward lipid metabolism, and the simultaneous repression of the class I and II HDACs (Tozzi et al., 2022). Furthermore, it has recently been shown that BHB particularly controls cellular processes including suppression of inflammation via a post-translational modification mechanism termed $\beta$-hydroxybutyrylation (Dąbek et al., 2020 ). Of note, despite their sharing the induction of increased ketone body content, the metabolic effects and outcomes at the central level of each ketogenic intervention differ. For example, compared with caloric restriction and exogenous ketone supplementation, only KDs have been observed to increase central levels of polyunsaturated fatty acids (e.g. docosahexaenoic acid) that exhibit anticonvulsant and neuroprotective properties (Bazan 2007; Fraser et al., 2003; Taha et al., 2005). Although the effects of KDs, exercise and fasting-related interventions including caloric restriction converge at counteracting mitochondrial dysfunction, involving the action of SIRT1 (Cantó et al., 2010; Maalouf et al., 2009), as well as prevention of apoptosis, the mechanisms inducing these effects may diverge (Maalouf et al., 2009). Once in the brain, ketones promote tissue integrity. The BHB-induced inhibition of class I and II HDACs increases the expression of the neurotrophin brain-derived neurotrophic factor (BDNF) (Koppel & Timmusk, 2013), as well as memory (Guan et al., 2009). BDNF is a small protein that acts on the membrane of neurons by binding to, and by activating, the membrane receptor tropomyosin-related kinase B (TrkB). To activate TrkB, the stored pro-BDNF pool is cleaved into mBDNF, triggered by excitatory synaptic activity. mBDNF is then secreted from the cell and binds to, phosphorylates and activates TrkB, inducing an intracellular signal transduction cascade involving CREB, Akt, nulcear factor-kappa B and activator protein-1, hence inducing its own expression and simultaneously increasing neuronal development, synaptic plasticity, cognition and energy homeostasis (Marosi & Mattson, 2014). The observations that (1) serum BHB and brain BDNF both increase upon intermittent fasting (Duan et al., 2001) and long-term aerobic exercise (Marosi et al., 2016; Sleiman et el., 2016; (2) BHB administration to the brain increases central BDNF levels (Sleiman et al., 2016); and (3) BHB induces BDNF expression in cortical neurons *in vitro* (Marosi et al., 2016) led to the suggestion that BHB triggers the upregulation of central BDNF levels in response to fasting and exercise (Duan et al., 2001; Marosi et al., 2016; Sleiman et al., 2016,). In analogy, KDs in rodents have been associated with increased serum BHB and brain BDNF (Acuña-Catalán et al., 2024; Di Lucente et al., 2024). BDNF has later been shown to also be expressed in peripheral tissues (Iu & Chan, 2022), including skeletal muscle in which its expression and activity is shown to be increased by fasting in mice (Yang et al., 2019), rats (Giacco et al., 2022) and humans (Walsh et al., 2015), as well as by voluntary wheel running in mice (Chan et al., 2024). Importantly, fasted muscle releases BDNF in the serum (Yang et al., 2019), which may be beneficial at the central level because BDNF is able to pass the blood–brain barrier (BBB) (Alcalà-Bazzara et al., 2010; Pan et al., 1998). Of note, also intense (non-ketogenic) exercise in humans has been shown to induce serum BDNF (Edman et al., 2024; Ferris et al., 2007; Gibbons et al., 2023; Kackley et al., 2022; Reycraft et al., 2020; Rojas Vega et al., 2006), associated with increased lactate rather than BHB levels. This specific non-ketogenic intervention type and the release of lactate should be elaborated separately and are not discussed further here. Finally, aerobically exercised muscle releases factors that influence cognitive function, including cathepsin B (Moon et al., 2016) and fibronectin type III domain-containing protein 5 (FNDC5)/irisin (Islam et al., 2021; Li et al., 2017; Lourenco et al., 2019). Given that (1) within muscle BHB uptake in response to fasting and exercise increases (Giacco et al., 2022; Kwak et al., 2021); (2) FNDC5/irisin expression responds to ketones (Kim et al., 2017); and (3) both cathepsin B (Moon et al., 2016) and irisin (Wrann et al., 2013) upregulate neuronal BDNF expression, skeletal muscle is emerging as an important mediator of cognition in response to ketogenic interventions.

The purpose of this review is to shed light on the link between ketogenic states induced by various physiological interventions and whether these induce improvement of cognition, discussing the interplay between organs, the bridging role of PPAR$\delta$ and the association with BDNF.

## From fatty acids to central action of ketones: roles of the PPARs

Induction of ketogenic states is well known to be the result of increased adipose lipolysis, fatty acid uptake in the liver, as well as their oxidation, and ketogenesis. These processes are governed by the PPAR superfamily of nuclear receptors. Natural fatty acids can be ligands of all three PPAR isoforms PPAR$\alpha$, $\gamma$ and $\delta$ although X-ray crystal-structure analysis revealed that ligand-binding pocket structures differ considerably among the various PPARs (Itoh et al. 2008; Xu et al., 1999), Ligand-binding assay studies showed that PPARs display the greatest preference for monounsaturated and polyunsaturated fatty acids (PUFAs) (Itoh et al. 2008; Krey et al., 1997). Fasting, a main inducer of the ketogenic state, is accompanied by increased serum free fatty acid (FFA) levels in humans (Barradas et al., 2022; Dobbins et al., 1998) and rats (Chasseigneaux et al., 2024; De Lange et al. 2006). This is a result of increased adipose lipolysis, which has been attributed to increased adipose PPAR$\delta$ activity in a study in mice (Yu et al., 2010). The resulting increase in PUFAs is sensed by the liver, which then mobilizes fatty acids and stimulates hepatic fatty acid oxidation (Clarke 2004) and ketogenesis (Desvergne & Wahli, 1999). Fatty acid sensing specifically involves PPAR$\delta$ based on a study by Sanderson et al. (2009), whereas PPAR$\alpha$ is the master transcriptional activator of ketogenic genes in the liver (Desvergne & Wahli, 1999). Other ketone-producing organs include the intestine (Bass et al., 2024), brain (Silva et al., 2022), kidneys (Packer, 2020) and adipose tissue (Nishitani et al., 2022), which are all discussed in the following section. One organ at the receiving end is skeletal muscle, which is able to switch fuel usage from glucose to fatty acids/ketones in response to energy deprivation, and PPAR$\delta$ has an important role in this process (De Lange et al., 2007). Indeed, fasting induces muscle PPAR$\delta$ mRNA levels in rodents (De Lange et al., 2006; Holst et al., 2003) but not in humans (Menezes et al., 2024; Wijngaarden et al., 2013), although, in humans, after a 3 day fast, increased capacity to oxidize lipids *in vitro* was demonstrated in permeabilized muscle fibres (Blackwood et al., 2023). Similarly, exercise increases skeletal muscle PPAR$\delta$ expression in mice (Chan et al., 2024; Chen et al., 2015; Luquet et al., 2003; Manio et al., 2018) and humans (Fritz et al., 2006; Greene et al., 2012; Mahoney et al., 2005; Menezes et al., 2024; Perry et al., 2010; Rundqvist et al., 2019). In response to aerobic exercise, PPAR$\delta$ regulates the expression of genes involved in muscle lipid content, $\beta$-oxidation and mitochondrial dynamics, thus contributing to muscle maintenance (Chan et al., 2024). Although not serving as metabolic fuel in the brain (Schonfeld & Reiser, 2017), fatty acids enter the brain through import within the BBB (Mitchell & Hatch, 2011). Of note, fatty acid subclasses including PUFAs possess neuroprotective activity (Bazan, 2007). Importantly, it has been shown that, as a result of the increased central influx of fatty acids, fasting-induced increased PPAR$\delta$ activity in the endothelial cells of the rat brain modifies BBB permeability by upregulating monocarboxylate transporter 1 (Slc16a1/MCT1) (Chasseigneaux et al., 2024). This finding is of importance because ketone entry into the brain via the BBB relies on MCT1 (Leino et al., 2001).

## Organ-specific ketone production and its modulation in response to various physiological interventions

**Liver.** During restricted availability of digestible carbohydrates (e.g. fasting, exercise), or in response to dietary interventions and supplements which stimulate ketone production, ketone bodies including BHB and acetoacetate are mainly produced by hepatic mitochondrial oxidation of FFA (Evans et al., 2017). Hydroxy-3-methylglutaryl-CoA synthase-2 (HMGCS2) mediates the rate-limiting step in mitochondrial ketogenesis. The liver produces and secretes ketone bodies and is unable to use ketones for energy production because it lacks succinyl-CoA:3oxoacid-CoA transferase, thus preventing ketolysis. A detailed review on the ketogenesis and ketolysis pathways is provided in Kolb et al. (2021).

**Intestine.** Fasting also induces ketone production in the intestine. It has recently been shown that the colon produces ketone bodies in response to fasting for maintenance of mitochondrial activity through the action of microbiota and that this leads to increased local and serum ketone levels because fasted conditional, colon-specific Hmgcs2-null mice had lower serum ketone levels (Bass et al., 2024).

**Brain.** It has recently been shown that the brain itself produces ketone bodies in response to fasting: A study on a 21 h period of food deprivation in *Drosophila melanogaster* revealed local ketone production in glial cells through this metabolic adaptation, which leads to memory improvement (Silva et al., 2022). The production and transfer of ketone bodies to neurons has been shown to depend on AMP-activated protein kinase (Silva et al., 2022).

**Other organs.** Furthermore, the kidneys produce ketones that protect mitochondrial function under pathological conditions including diabetes and chronic kidney disease (Packer, 2020). Of note, kidney-induced ketogenesis does not contribute to circulating ketones (Venable et al., 2022). Finally, white adipocytes may produce and excrete BHB levels: it has recently been shown that HMGCS2 is expressed in epidydimal adipose tissue, and the induction

of 3T3-L1 adipocyte differentiation increases HMGCS2 expression and BHB production and secretion (Nishitani et al., 2022). The contribution of adipose ketogenesis to serum BHB levels *in vivo* and the effect of physiological interventions has not yet been established. Interestingly, one study (Tsai et al., 2022), found that raspberry ketones induced adipocyte expression of FNDC5/irisin, known to exert positive effects on cognition (see the following section) (Islam et al., 2021; Li et al., 2017; Lourenco et al., 2019), which could indicate that the adipose ketogenic state contributes to irisin production. Further research is warranted to bring these observations into an *in vivo* context. Figure 1 provides an overview of the organs that produce ketones and their response to physiological interventions, resulting in increased local and, in specific cases, serum ketone levels.

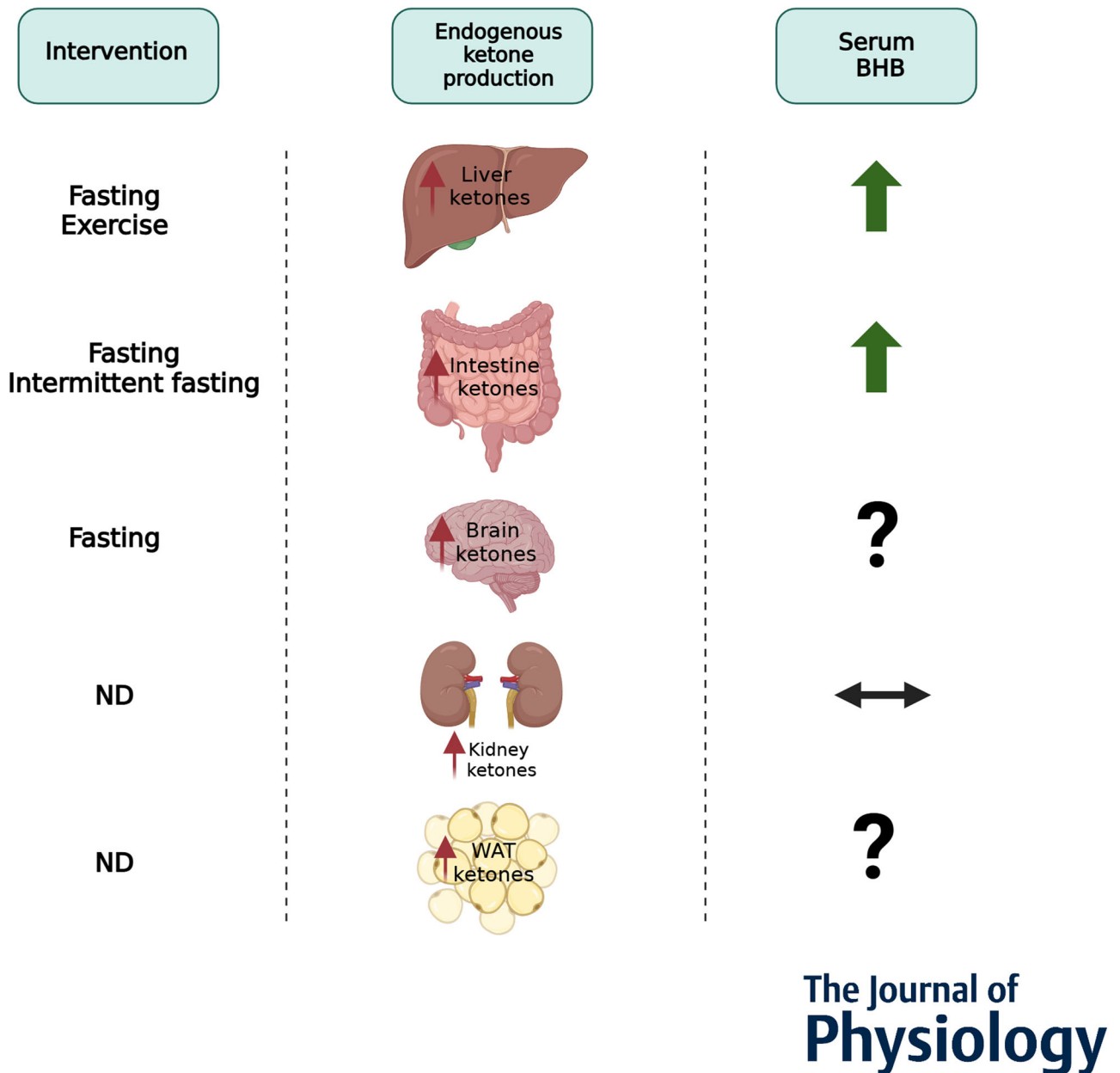

**Figure 1. Overview of the ketone-producing organs**
The ketone-producing organs and the modulatory effects, where studied, of exercise, fasting, intermittent fasting and the eventual resulting increase in serum BHB levels. Abbreviation: ND, Not determined (no physiological interventions reported in the literature)

## Central effects of ketogenic interventions through the intestine and skeletal muscle

**Re-shaping of the microbiome in response to ketones: beneficial central outcomes.** Although local production of ketone bodies may shape the gut microbiome (Ang et al., 2020; Bass et al., 2024), it is conceivable that further elevated serum levels of hepatic BHB in response to KD or similar interventions may directly or indirectly affect the gut microbiota (Ang et al., 2020). Furthermore, the way in which the ketogenic state is induced may determine the shaping of the microbiome. Indeed, although, in humans, KD has been reported to decrease the abundance of *Bifidobacterium* (Ang et al., 2020), interestingly, calorie restriction, but not intermittent fasting, has been reported to increase *Bifidobacterium* abundance in female mice (Mao et al., 2023). Interestingly, in a clinical trial on elderly twin pairs submitted to resistance exercise and supplementation with branched-chain amino acids, a prebiotic supplement that increased *Bifidobacterium* abundance increased cognitive ability (Ni Lochlain et al., 2024). Fasting in rats (Giacco et al., 2020) as well as KD in mice (Olson et al., 2018) were demonstrated to increase the abundance of Parabacterioides, which has been shown to have a beneficial effect on epileptic seizures when fed by oral gavage in antibiotic-treated mice fed a control diet (Olson et al., 2018). Besides ketone bodies, the structurally related short-chain fatty acid (SCFA) butyrate and other SCFAs (e.g. acetate, propionate) are key products of microbiota. Each of these metabolites show promising effects in neurological disorders (Stilling et al., 2016). Indeed, the KD-induced alteration of the microbiota profile has been observed to change the colonic lumenal, serum and hippocampal metabolomic profiles that correlate with seizure protection (Olson et al., 2018). Modulation of the gut microbiome by a KD has been shown to ameliorate Alzheimer's disease markers in human subjects with mild cognitive impairment (Nagpal et al., 2019). Of note, one study that compared the effects of KD and intermittent fasting on beneficial outcomes of Alzheimer's disease in rats questioned the direct involvement of serum ketones themselves. Both interventions increased serum ketone levels but shaped the microbiome in a differential manner and only intermittent fasting was beneficial for hippocampal amyloid-$\beta$ deposition and memory function (Park et al., 2020). From these findings, it may be concluded that the way ketogenic states are induced brings about different microbiome signatures with specific metabolites, with different central outcomes.

**Skeletal muscle: a contributor in the mediation of ketone-induced centrally beneficial effects.** Skeletal muscle uses ketone bodies derived from the liver as fuel in ketogenic conditions such as fasting (Anton et al., 2018; Evans et al., 2017) and, indeed, BHB levels have been shown to be increased within rodent muscle tissue during fasting (Giacco et al., 2022) and mild endurance exercise (Giacco et al. 2022; Kwak et al., 2021), but not resistance (non-ketogenic) exercise (Kwak et al., 2021). Mild endurance exercise in combination with fasting has been shown to further increase BHB levels in muscle (Giacco et al., 2022). Ketones are known to preserve muscle mass and function during fasting and exercise (Anton et al., 2018). Similarly, KDs are beneficial for conserving muscle mass during ageing (Pathak & Baar, 2023). The monocarboxylate transporters MCT1-8, involved transport of compounds including ketones (Bonen et al., 2006), have all found to be expressed in rat as well as human skeletal muscle (Bonen et al., 2006). In particular, monocarboxylate transporter 2 (SLC16A7/MCT2) rather than MCT1 (SLC16A1/MCT1) has been shown to be upregulated in muscle in response to fasting in rats (Giacco et al., 2022) and exercise in rats (Giacco et al., 2022) and mice (Béland-Millar et al., 2020). Ketone metabolism in muscle generates ATP as a result of their conversion into acetyl coenzyme A, which then enters the tricarboxylic acid cycle (Cahill, 2006). In addition, ketone metabolism preserves muscle protein content (Evans et al., 2017), reflected by reduced alanine release during starvation (Sherwin et al. 1975), as well as reduced leucine oxidation (Nair et al. 1988). The metabolic effects of ketone uptake thus guarantee maintenance of muscle integrity. In response to exercise, skeletal muscle secretes many factors, termed myokines, which allow cross-talk with the brain (Townsend et al., 2021). Several myokines influence cognitive function, including cathepsin B (Moon et al., 2016) and FNDC5/irisin (Islam et al., 2021; Li et al., 2017; Lourenco et al., 2019). Importantly, ageing-related low skeletal muscle mass in humans has been associated with reduced cognition through decreased mitochondrial function and dysfunctional myokine secretion, with physical activity being proposed to restore myokine balance (Oudbier et al., 2022). A direct link between increased muscle BHB in response to fasting and exercise (Giacco et al., 2022; Kwak et al., 2021) and increased FNDC5/irisin expression has been discovered because treatment with the histone deacetylase inhibitor and ketone precursor sodium butyrate increased mRNA expression of FNDC5 in Huh7 cells (Kim et al., 2017). However, in male rats, serum irisin levels have been shown to be unaltered in response to a 48 h fast (Quiñones et al., 2015). Interestingly, mild exercise has been shown to increase muscle-specific thyroid hormone (T3) production and, subsequently, to increase serum and prefrontal cortex levels of fasted rats (Giacco et al., 2022). There was a significant response in the expression of prefrontal cortex genes involved in neuronal regeneration known to be under the transcriptional control of T3 (Giacco et al., 2022). Aerobic exercise has been found to increase serum T3 and T4 levels (Altaye et al., 2019;

Ciloglu et al., 2005), which have been shown to correlate with academic improvement in students (Altaye et al., 2019). Confirmation of a direct effect of thyroid hormone on cognition comes from a study with hypothyroid mice, in which T4 (levothyroxine) treatment rescued impaired performance in novel object recognition, and the 20% reduction of neuroprogenitor numbers in hippocampal neurogenic niches (Rutigliano et al., 2023). In this context, it is worth mentioning that treatment of thyroidectomized patients with levothyroxine (T4) has been shown to increase serum irisin levels which directly correlated with circulating free T3 (fT3) (Bocale et al., 2021). Interestingly, treatment of male, but not female, rats with the serotonin reuptake inhibitor paroxetine, a drug used in the treatment of mood disorders, showed an increase in serum fT3, fT4 and thyroid-stimulating hormone levels in response to simultaneous treatment with irisin (Ercan et al., 2023). Importantly, protein levels of the neuroprotective factor BDNF in muscle are related to the ketogenic state induced by fasting (Giacco et al., 2022; Walsh et al., 2015; Yang et al., 2019) and aerobic exercise (Chan et al., 2024) and this factor is emerging as a myokine. This is discussed in detail in the final section of this review.

## The link between ketones, brain BDNF and cognition

Increased cognition upon influx of ketones in the brain is thought to depend on increased activity of BDNF (Fig. 2). The observation that BDNF induces neuronal development, synaptic plasticity and cognition in response to energetic challenges (Marosi & Mattson, 2014) led to numerous studies investigating the link between BHB, BDNF and cognition in response to various ketogenic interventions, which will be addressed in this section.

**Exogenous BHB addition and serum and brain BDNF.** In cultured mouse cerebral cortical neurons (Marosi et al., 2016; Sleiman et al., 2016) and on mouse hippocampal slices (Sleiman et al., 2016), BHB has been shown to increase BDNF expression, and injection of BHB into the mouse brain increased hippocampal BDNF expression (Sleiman et al., 2016). In addition, BHB administration to glial cells increased BDNF mRNA (Kwak et al., 2021). Exogenous administration of the BHB precursor 1,3 butanediol to rats also increased BDNF protein levels in the hippocampus (Cigliano et al., 2025). In a mouse model of Alzheimer's disease, BHB administration prevented the fall in long-term potentiation in the hippocampus (Di Lucente et al., 2024). Beneficial effects of exogenous administration of BHB in humans have been found in subjects with type 2 diabetes showing increased working memory performance upon BHB infusion (Jensen et al.,

2020). Of note, low BDNF serum levels are linked to depression both in mice (Wei et al., 2023) and humans (Sato et al., 2023). A correlation between BHB and serum BDNF in humans is emerging because a ketone supplement has been shown to prevent suppression of circulating BDNF upon ingestion of carbohydrates in lean individuals (Walsh et al., 2020), and ingestion of an oil containing BHB has been shown to increase serum proBDNF in mixed-sex healthy volunteers (Norgren et al., 2021).

**KDs: central effects in rodents and humans, toward a link between BHB and BDNF.** A 7 month KD in a mouse model of Alzheimer's disease rescued long-term potentiation in the hippocampus to wild-type levels without altering amyloid-$\beta$ levels (Di Lucente et al., 2024). The KD significantly induced synaptic plasticity, phosphorylated-extracellular signal regulated kinase and phosphorylated-CREB enzymes in both sexes and of brain BDNF only in female mice (Di Lucente et al., 2024). In apparent contrast, the effect of a short-term KD on aged male mice, namely improving working memory and hippocampal long-term potentiation through protein kinase A-induced synaptic plasticity, did result in high hippocampal BDNF abundance (Acuña-Catalán et al., 2024). In both studies, serum BHB was reported to be increased (Acuña-Catalán et al., 2024; Di Lucente et al., 2024) Of note, a KD in humans resulted in a four-fold increase in serum BHB levels and a two-fold increase in serum FFA concentrations (Zajac et al., 2014). Similarly, a 2–3 month exposure to a very low carbohydrate diet in obese subjects has been shown to increase both serum irisin and BHB levels, whereas a low calory diet did not (Sajoux et al., 2019).

**Fasting, ketones and BDNF in serum and brain.** Acute fasting (9 h) in mice has been shown to induce BDNF expression in the cortex and hippocampus (Cui et al., 2018), but BHB levels have not been measured. In apparent contrast, despite increases in serum BHB (Giacco et al., 2022), no increase in BHB and BDNF protein nor in the BNDF-TrkB-Akt-CREB pathway was observed in prefrontal cortex, of rats after a 66 h fast (Giacco et al., 2022). In agreement, 48 h fasting in mice did not increase hypothalamic BDNF protein (Gilland and Fox, 2017). However, longer-term exposure of rats to ketones to intermittent fasting for a period of 3 months has been shown to increase BDNF protein levels in the hippocampus, cortex and striatum (Duan et al., 2001), and a 12 week period of caloric restriction in mice increased serum BHB and hippocampal BDNF in diabetic (db/db) mice (Stranahan et al., 2009). Although a human study has reported an increase of both serum and brain BHB upon a 2 day fast (Pan et al., 2000), one reason why BHB induced

by fasting is not always effective as a signalling molecule at the central level may be that it is metabolized acutely (Cahill, 2006). It is conceivable that intermittent fasting and caloric restriction may depend to a lesser extent on the immediate central use of BHB as fuel. In human studies, serum BDNF levels have been measured as a potential marker of beneficial effects of fasting-induced ketogenesis, with varying results: a 20 h fast induced serum BHB without resulting in increased serum BDNF levels in humans (Gibbons et al., 2023), whereas, in contrast, a

3 day fast has been shown to induce both serum BHB and BDNF in humans (Edman et al., 2024).

**Exercise under fed and energy-restricted conditions, ketones and BDNF in serum and brain.** In rodents, short bouts of moderate-to-intense exercise have been shown to increase BDNF-TrkB signalling in the pre-frontal cortex (Baranowski and MacPherson, 2018; Cefis et al., 2019) and hippocampus (Cefis et al., 2019).

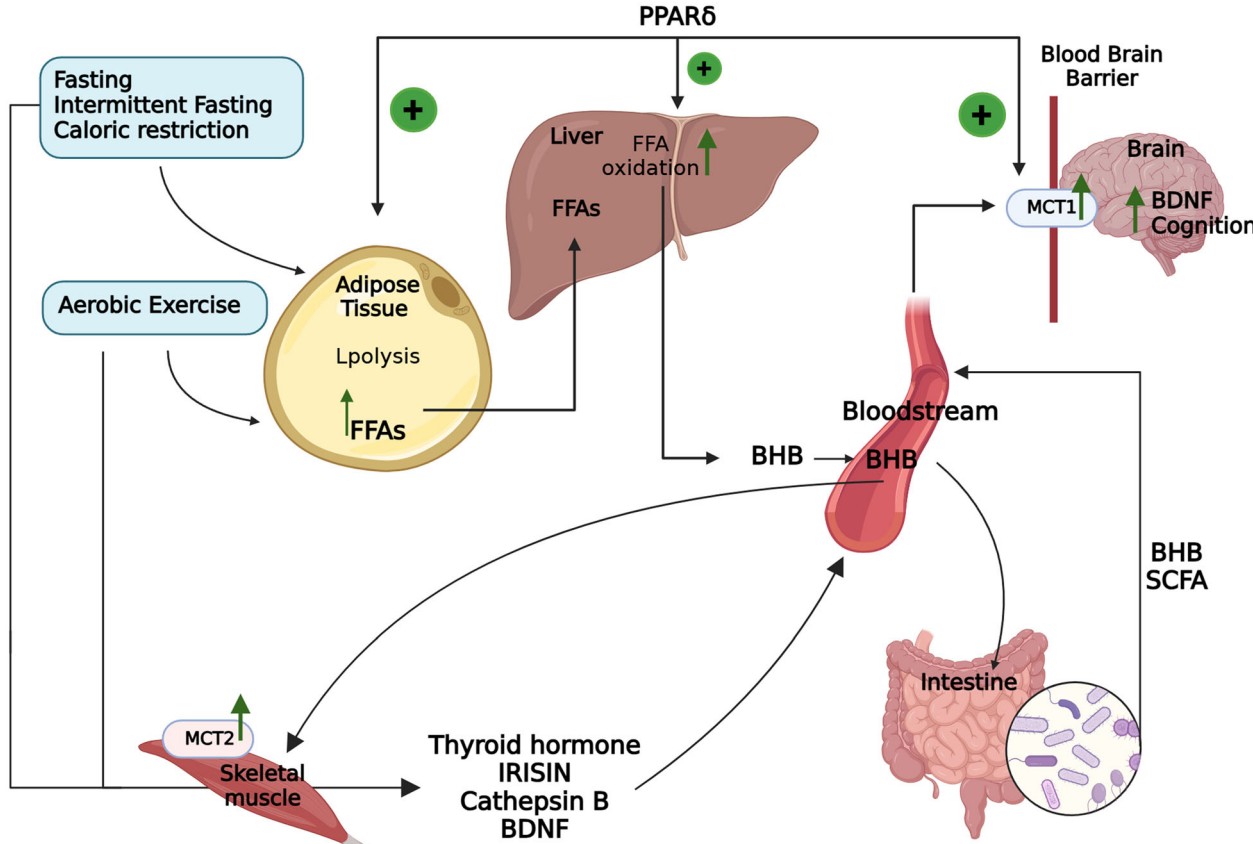

**Figure 2. Overview of events and organs in fasting- and exercise-induced cognition**
The sequence of events and the organs involved that lead to fasting- and exercise-induced cognition, with a regulatory role for PPARδ spanning from adipose FFA release to cerebral influx of ketones, as well as muscle-derived factors and metabolites related to altered microbiota signatures

BHB serum levels have not been measured in these studies. Repeated short-term mild endurance exercise (five 20 min low-speed treadmill running sessions twice a day for 66 h) in rats was found to increase BDNF-TRKB-Akt-CREB signalling in prefrontal cortex; however, serum and prefrontal cortex BHB levels did not increase (Giacco et al., 2020, 2022). The same exercise intervention under fasting conditions failed to induce pre-frontal cortex BDNF-TRKB-Akt-CREB signalling, again with no increase in prefrontal cortex tissue BHB levels (Giacco et al., 2022), although serum BHB levels were increased (Giacco et al., 2020). A study on voluntary wheel running for 30 days in mice showed a correlation between increased serum BHB and hippocampal BDNF (Sleiman et al., 2016). The same results were obtained after 6 weeks of voluntary wheel running in mice (Marosi et al., 2016). Long-term mild exercise (4–6 min of treadmill running for 5 weeks) in rats has been reported to reverse age-related impairments in spatial learning and long-term memory. This was accompanied by an increase in hippocampal BDNF-Akt-CREB signalling (Aguiar et al., 2011). Long-term mild exercise (30 min of low-speed daily treadmill running for 1 month) in Parkinsonian mice has recently been shown to counter-act α-synuclein spreading and to prevent early synaptic deficits, ameliorating motor and corticostriatal long-term potentiation. These processes have been shown to involve increased BDNF-TrkB signalling in the rescue of striatal synaptic plasticity, indicating a possible mechanism for the recovery of motor and visuospatial behaviour (Marino et al., 2023). In the latter two rodent studies, serum and tissue BHB levels were not measured, although, based on the two above-mentioned studies (Marosi et al., 2016; Sleiman et al., 2016), serum BHB levels may have increased during these long-term, moderate inter-ventions. One study that does not confirm the previous findings (Marosi et al., 2016; Sleiman et al., 2016) regards a 12 week period of voluntary wheel running in diabetic (db/db) mice (Stranahan et al., 2009). Serum BHB levels did not increase, although hippocampal BDNF levels did increase. Increased serum BHB levels in response to a simultaneous caloric restriction intervention were not altered with exercise, whereas hippocampal BDNF levels remained increased (Stranahan et al., 2009). In analogy, a 90 min, low-intensity exercise session during a 20 h fasting period in humans did not result in an increase in serum BHB levels over those induced by fasting alone (Gibbons et al., 2023). Serum BDNF levels did not increase compared to sedentary fed controls in this study (Gibbons et al., 2023). The discrepancy between the above findings indicates that further research is necessary to increase our understanding regarding the association between mild exercise, the ketogenic state and central action of BDNF. An overview of the discussed findings is presented in Table 1.

**Towards a link between BHB, muscle, BDNF, and cognition.** As mentioned earlier, reduced alanine release (Sherwin et al., 1975) and reduced leucine oxidation (Nair et al., 1988) comprise a measure of preserved skeletal muscle mass in response to ketones. Loss of skeletal muscle mass in ageing humans has been associated with reduced mitochondrial function and myokine secretion and, importantly, this is associated with decreased BDNF content, with physical exercise being suggested to counteract these events (Oudbier et al., 2022). In line with this, recently, voluntary wheel running, known to increase serum BHB in mice (Marosi et al., 2016; Sleiman et al., 2016), has been shown to increase muscle BDNF in mice (Chan et al., 2024). Interestingly, increased expression of muscle PPARδ and its target genes in response to low-intense treadmill running (Chan et al., 2024) was shown to depend on BDNF because muscle-specific BDNF knockout mice exhibited impairments in the expression of PPARδ and PPARδ-regulated genes involved in muscle lipid content, fatty acid oxidation and mitochondrial dynamics (Chan et al., 2024). Taken together, these data support an inter-action between BDNF and PPARδ in the link between ketone- and fatty acid-induced muscle health, as well as cognition. In association with increased BHB tissue levels in fasting rat muscle (Giacco et al., 2022), skeletal muscle BDNF expression has been shown to be increased during fasting in mice (Yang et al., 2019), rats (Giacco et al., 2022) and humans (Walsh et al., 2015). A 3 day fast has been shown to induce both serum BHB and BDNF in humans (Edman et al., 2024). Importantly, the amount of circulating BDNF in response to fasting has been shown to be significantly reduced in muscle-specific BDNF knockout mice (Yang et al., 2019), providing conformation that a significant part of circulating BDNF in the fasting state is released from muscle. A second study also reported reduced serum BDNF levels in muscle-specific BDNF knockout mice (Fulgenzi et al., 2020). In the same study, BDNF was shown to be secreted from differentiated human primary myoblasts and to be responsible for inducing insulin release from human pancreatic islets (Fulgenzi et al., 2020). The discovery that serum-derived BDNF can enter the CNS upon transport through the BBB (Alcalà-Bazzara et al., 2010; Pan et al., 1998) renders muscle-secreted BDNF in the ketogenic state a candidate for improved cognition. Adding to evidence suggesting a link between ketones and irisin (Kim et al., 2017), evidence also exists for an association between cathepsin and irisin, and central BDNF activation. Indeed, cathepsin B, upon addition to neuronal precursor cells *in vitro*, has been shown to induce BDNF expression (Moon et al., 2016) and irisin was found to induce the expression of BDNF in the hippocampus in response to prolonged mild exercise (Wrann et al., 2013). Finally, in line with the observation that thyroid hormone following exercise in

**Table 1. Relationship between different interventions, BHB, T3, BDNF and cognition in rodents and humans.**

| Intervention | BHB serum | BHB brain | T3 serum | T3 brain | BDNF cells | BDNF brain | BDNF serum | Cognition | Epileptic seizure inhibition | References |
|---|---|---|---|---|---|---|---|---|---|---|
| Incubation of cortical neurons with BHB | ND | ND | ND | ND | + (a) | ND | ND | ND | ND | Marosi et al. (2016); Sleiman et al. (2016) |
| Incubation glial cells with BHB | ND | ND | ND | ND | + (a) | ND | ND | ND | ND | Kwak et al. (2021) |
| Incubation BHB hippocampal slices | ND | + (a) | ND | ND | ND | + (a) | ND | ND | ND | Sleiman et al. (2016) |
| Incubation BHB hippocampal slices | ND | ND | ND | ND | ND | ND | ND | + (a) | ND | Di Lucente et al. (2024) |
| Injection BHB into brain | ND | ND | ND | ND | ND | + hippocampus (a) | ND | ND | ND | Sleiman et al. (2016) |
| BHB infusion | ND | ND | ND | ND | ND | ND | ND | + (h) | ND | Jensen et al. (2020) |
| Ingestion of ketone supplement | ND | ND | ND | ND | ND | ND | + (h) | ND | ND | Walsh et al. (2020) |
| BHB administration | ND | ND | ND | ND | ND | ND | ND | + (a) | ND | Di Lucente et al. (2024) |
| BHB precursor administration | + | ND | ND | ND | ND | + hippocampus (a) | ND | ND | ND | Cigliano et al. (2025) |
| BHB in oil ingested | ND | ND | ND | ND | ND | ND | + (h) | ND | ND | Norgren et al. (2021) |
| Ketogenic diet | + (a) | ND | ND | ND | ND | + hippocampus (a) | ND | + (a) | ND | Di Lucente et al. (2024) |
| Ketogenic diet | + (a) | ND | ND | ND | ND | + hippocampus (a) | ND | + (a) | ND | Acuña-Catalán et al. (2024) |
| Ketogenic diet | + (h) | ND | ND | ND | ND | ND | ND | ND | ND | Zajac et al. (2014) |
| Ketogenic diet | ND | ND | ND | ND | ND | ND | ND | + (h) | ND | Nagpal et al. (2019) |
| Ketogenic diet | ND | ND | ND | ND | ND | ND | ND | – (a) | ND | Park et al. (2020) |
| Ketogenic diet, increased parabacteroides abundance | ND | ND | ND | ND | ND | ND | ND | ND | + (a) | Olson et al. (2018) |
| Fasting, 9 h | ND | ND | ND | ND | ND | +cortex hippo campus (a) | ND | ND | ND | Cui et al. (2018) |

*(Continued)*

**Table 1. (Continued)**

| Intervention | BHB serum | BHB brain | T3 serum | T3 brain | BDNF cells | BDNF brain | BDNF serum | Cognition | Epileptic seizure inhibition | References |
|---|---|---|---|---|---|---|---|---|---|---|
| Fasting, 20 h | + (h) | ND | ND | ND | ND | ND | – (h) | ND | ND | Gibbons et al. (2023) |
| Fasting, 21 h | ND | + (a) | ND | ND | ND | ND | ND | + (a) | ND | Silva et al. (2022) |
| Fasting, 48 h | ND | ND | ND | ND | ND | –hypothalamus (a) | ND | ND | ND | Gilland and Fox (2017) |
| Fasting, 66 h | + (a) | ND | ND | ND | ND | ND | ND | ND | ND | Giacco et al. (2020) |
| Fasting, 66 h | ND | – (a) | – (a) | – (a) | ND | – cortex (a) | ND | ND | ND | Giacco et al. (2022) |
| Fasting, 2 days | + (h) | + occipital lobe (h) | ND | ND | ND | ND | ND | ND | ND | Pan et al. (2000) |
| Fasting, 3 days | + (h) | ND | ND | ND | ND | ND | + (h) | ND | ND | Edman et al. (2024) |
| Intermittent fasting, 3 months | ND | ND | ND | ND | ND | + hippocampus, cortex, striatum (a) | ND | ND | ND | Duan et al. (2001) |
| Intermittent fasting, 8 weeks | ND | ND | ND | ND | ND | ND | ND | + (a) | ND | Park et al. (2020) |
| Caloric restriction, 12 weeks | + (a) | ND | ND | ND | ND | + hippocampus (a) | ND | ND | ND | Stranahan et al. (2009) |
| Exercise, short | ND | ND | + (h) | ND | ND | ND | ND | + (h) | ND | Altaye et al. (2019) |
| Exercise, short | ND | ND | ND | ND | ND | + cortex (a) | ND | ND | ND | Baranowski and MacPherson (2018) |
| Exercise, short | ND | ND | ND | ND | ND | + cortex (a) / + hippocampus (a) | ND | – | ND | Cefis et al. (2019) |
| Exercise, daily short treadmill running, 5 weeks | ND | ND | ND | ND | ND | + hippocampus (a) | ND | + (a) | ND | Aguiar et al. (2011) |
| Exercise voluntary wheel running, 30 days | + | ND | ND | ND | ND | + hippocampus (a) | ND | ND | ND | Sleiman et al. (2016) |
| Exercise voluntary wheel running, 6 weeks | + | ND | ND | ND | ND | + hippocampus (a) | ND | ND | ND | Marosi et al. (2016) |

*(Continued)*

**Table 1. (Continued)**

| Intervention | BHB serum | BHB brain | T3 serum | T3 brain | BDNF cells | BDNF brain | BDNF serum | Cognition | Epileptic seizure inhibition | References |
|---|---|---|---|---|---|---|---|---|---|---|
| Mild treadmill exercise, 30 days | ND | ND | ND | ND | ND | +striatum (a) | ND | + (a) | ND | Marino et al. (2023) |
| Repeated mild exercise, short | − (a) | ND | ND | ND | ND | ND | ND | ND | ND | Giacco et al. (2020) |
| Repeated mild exercise, short | ND | − (a) | − (a) | − (a) | ND | +cortex (a) | ND | ND | ND | Giacco et al. (2022) |
| Fasting 66 h. Repeated mild exercise, short | + (a) | ND | ND | ND | ND | ND | ND | ND | ND | Giacco et al. (2020) |
| Fasting 66 h. Repeated mild exercise, short | ND | − (a) | + (a) | + (a) | ND | − cortex (a) | ND | ND | ND | Giacco et al. (2022) |
| Exercise low intensity, 90 min fasting | + (h) | ND | ND | ND | ND | ND | − (h) | ND | ND | Gibbons et al. (2023) |
| Exercise voluntary wheel running, 12 weeks | − (a) | ND | ND | ND | ND | + hippocampus (a) | ND | ND | ND | Stranahan et al. (2009) |
| Exercise, intense wheel running, 3 weeks | − (a) | ND | ND | ND | ND | ND | ND | ND | ND | Chen et al. (2015) |
| Exercise voluntary wheel running, 14 days | ND | ND | ND | ND | ND | + hippocampus (a) | ND | ND | ND | Wrann et al. (2013) |
| Exercise and caloric restriction | + (a) | ND | ND | ND | ND | + hippocampus (a) | ND | ND | ND | Stranahan et al. (2009) |
| Exercise short intense | ND | ND | ND | ND | ND | + cortex (a) | ND | + (a) | ND | Cefis et al. (2019) |
| Resistance exercise +prebiotic supplement | ND | ND | ND | ND | ND | ND | ND | + (h) | ND | Ni Lochlain et al. (2024) |
| Levothyroxine | ND | ND | + (h) | ND | ND | ND | ND | ND | ND | Bocale et al. (2021) |
| Levothyroxine, hypothyroid mice | ND | ND | − (a) | ND | ND | + hippocampus (a) | ND | + (a) | ND | Rutigliano et al. (2023) |
| Levothyroxine and exercise | ND | ND | ND | ND | ND | + hippocampus (a) | ND | + (a) | ND | Boustani et al. (2024) |

Abbreviations: a, animals; h, humans; +, increase; −, no change; ND, not determined. For additional information, see text.

the ketogenic state is released from skeletal muscle and accumulates in the brain (Giacco et al., 2022), it has recently been shown that levothyroxine (T4) treatment in hypothyroid male offspring rats submitted to mild exercise ameliorates deficits of spatial navigation, as well as the anxiety profile, and increases hippocampal BDNF (Boustani et al., 2024). Further studies are warranted to shed light on the link between these mild exercise-induced factors that are released from muscle in ketogenic states and the induction of beneficial central effects through BDNF.

An overview is provided in Fig. 2 of the known effects of ketogenic interventions on (1) the flow of FFA from the adipose tissue to the liver; (2) subsequent ketone pools to muscle and the intestine as well as through the BBB; and (3) the resulting flow of factors increasing cognition to the brain.

**General conclusions.** Ketogenesis through physiological interventions is essential for cerebral metabolism and neurogenesis, with the flow of ketones from the periphery to the brain being under control of PPAR$\delta$. Although ketones themselves relate to cognition, it has to be taken into account that additional factors induced in tissues, including the skeletal muscle and the intestine, by the moderate physiological interventions discussed in this review may have similar effects. It is important to investigate each intervention separately to obtain a clear insight into the signals that are involved in improvement of cognition. It may be concluded that physiological approaches that induce a ketogenic state and modulate metabolism can improve cognition, which needs to be further explored in the future.

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

## Additional information

### Competing interests

The authors declare that they have no competing interests.

### Author contributions

A.G. drafted the main part of the paper. G.P. drafted parts of the paper and designed figures. R.S. drafted parts of the paper. M.M., A.Lo., and A.La. edited the paper and modified parts. Pd.L. created the concept and wrote and edited the final version. All authors approved the final version of the manuscript submitted for publication. All persons designated as authors qualify for authorship, and all those who qualify for authorship are listed.

### Funding

We acknowledge financial support under the National Recovery and Resilience Plan (NRRP), Mission 4, Component 2, Investment 1.1, Call for tender No. 1409 published on 14.9.2022 by the Italian Ministry of University and Research (MUR), funded by the European Union – NextGenerationEU – Project Title Mechanisms underlying the beneficial activities of ketogenic states in healthy and epileptic conditions: insights extended to a genetically modified mouse model – CUP University of Campania Luigi Vanvitelli B53D23033340001 (Pieter de Lange), CUP University of Naples Federico II E53D23021790001 (Assunta Lombardi)- Grant Assignment Decree No. 1363 adopted on September 1st 2023 by the Italian Ministry of University and Research (MUR).

### Keywords

BDNF, cognition, exercise, fasting, ketogenic diet, ketones

### Supporting information

Additional supporting information can be found online in the Supporting Information section at the end of the HTML view of the article. Supporting information files available:

**Peer Review History**

