## [Peer Review History · The Journal of Physiology]

The Central Benefit of Physiologically Induced Ketogenic States

Antonia Giacco, Giuseppe Petito, Rosalba Senese, Maria Moreno, Assunta Lombardi, Antonia Lanni, and Pieter de Lange
DOI: 10.1113/JP287462

Corresponding author(s): Pieter de Lange (pieter.delange@unicampania.it)

Review Timeline:

Submission Date:	14-Oct-2024
Editorial Decision:	09-Jan-2025
Revision Received:	30-Jan-2025
Editorial Decision:	05-Mar-2025
Revision Received:	17-Mar-2025
Accepted:	03-Apr-2025

Senior Editor: Karyn Hamilton

Reviewing Editor: Val Andrew Fajardo

Transaction Report:

Dear Dr de Lange,

Re: JP-TR-2024-287462 "The Central Benefit of Physiologically Induced Ketogenic States" by Antonia Giacco, Giuseppe Petito, Rosalba Senese, Maria Moreno, Assunta Lombardi, Antonia Lanni, and Pieter de Lange

Thank you for submitting your manuscript to The Journal of Physiology. It has been assessed by a Reviewing Editor and by 1 expert referee(s) and we are pleased to tell you that it is potentially acceptable for publication following satisfactory major revision.

Please address all the points raised and incorporate all requested revisions or explain in your Response to Referees why a change has not been made. We hope you will find the comments helpful and that you will be able to return your revised manuscript within 9 months. If your article is for a Special Issue, please note that we require your revised version within 2 months (rather than 9 months) in order to keep the Special issue on track. If you require longer than this, please contact journal staff: jp@physoc.org. Please note that this letter does not constitute a guarantee for acceptance of your revised manuscript.

ABSTRACT FIGURES: Authors are expected to use The Journal's premium BioRender account to create/redraw their Abstract Figures. Information on how to access this account is here:

<https://physoc.onlinelibrary.wiley.com/journal/14697793/biorender-access>.

REVISION CHECKLIST:

IMPORTANT POINTS TO NOTE WHEN REVISING YOUR MANUSCRIPT:

We look forward to receiving your revised submission.
If you have any queries, please reply to this email and we will be pleased to advise.

Yours sincerely,

Karyn Hamilton
Senior Editor
The Journal of Physiology

EDITOR COMMENTS

Reviewing Editor:

Comments to the Author:

The report from the reviewer indicates that this topic is timely and the review paper has the potential to be influential. However, the reviewer raised several points that need to be addressed in addition to recommending that the review paper be edited to enhance the flow and/or presentation of key concepts.

Senior Editor:

Comments to the Author:

Thank you for submitting your Topical Review for consideration in this Special Issue of The Journal of Physiology. We apologize for the longer than ideal time that lapsed since you submitted the manuscript--the holiday season is a particularly difficult time to secure peer Referees. The Referee whose feedback we secured believed that you've revised an important and interesting topic that is relevant to the field of research. However, the Referee and our editorial team believe that major revisions would be necessary for the manuscript to be further considered for publication. These include (but are not limited to) 1. reworking the introduction to appropriately set up the purpose of the review paper, 2. revisiting several sections that currently lack focus and provide only surface level information, 3. revising table information, and 4. overall improving the flow of the review. We would like to invite you to respond to and address each Referee critique with a revised manuscript. We look forward to seeing your revised review if you choose to submit one. Thank you again for your interest in The Journal of Physiology.

REFEREE COMMENTS

Referee #2:

In this review paper, Giacco and colleagues review the central benefits of varying ketogenic states and discuss potential mediators behind these effects, such as BDNF and PPAR delta. This review covers an important and interesting topic that is relevant to the field of research. Although the authors present good data, several major revisions are required before this review is ready for publication. Overall, this review paper was difficult to follow along to, lacks focus and is missing adequate flow.

Major points

- 1) The information provided in the introduction does not appropriately set up the purpose of the review paper.
- Surface level information is provided as to why Ketones are of interest
 - Further information on BDNF is required, expand on the connection between BDNF and Ketones and more so why we care about increasing BDNF levels
 - Exercise and fasting are major components in this review paper yet little to no information is provided in the introduction
 - No information on PPAR delta is provided in the introduction

2) "Physiological interventions modulating the ketogenic state" section lacks focus

- The authors spend more time discussing organs that produce ketones rather than the actual physiological interventions that modulate a ketogenic state.
- The authors should make a figure for the different organs that produce ketones and use this section to discuss how exercise, fasting and supplementation leads to increased ketone production

3) "Central responses to ketones, lactate or both" section provides surface level information and is scattered.

- The headings are difficult to follow along with, the authors jump from point to point with a lack of flow
- This is the first time the authors bring up lactate, what is the relationship between BHB and lactate? Despite lactate being an interesting topic, it does not fit within the scope of the authors review topic and should be removed.
- Why do the authors only discuss a ketogenic diet in rodents? what about humans?

4) Table 1 is difficult to read and not of publication quality, authors should find a better way to present this data.

5) The review, after table 1, becomes heavily about PPAR delta, a lot of the information provided in this section should have been brought up in the introduction of the review. The depth of information in this section makes the reviews appear to be primarily on PPAR delta rather than the central benefits of ketogenic states.

6) The addition of the FGF21 discussion at the end of the review seems unnecessary and lacks relevance to the rest of the review. The authors are urged to narrow the scope of the review to what is directly relevant to the topic.

END OF COMMENTS

Response to Referees Giacco et al.

We would like to thank the Senior and Reviewing Editor as well as referee #2 for their constructive comments. Below, please find the point-to-point replies to their reports, with the responses being highlighted in colour.

EDITOR COMMENTS

Reviewing Editor:

Comments to the Author:

The report from the reviewer indicates that this topic is timely and the review paper has the potential to be influential. However, the reviewer raised several points that need to be addressed in addition to recommending that the review paper be edited to enhance the flow and/or presentation of key concepts.

We thank the Reviewing Editor for his/her comments. We have taken all the comments and suggestions of referee #2 into account, hoping this version of the manuscript sufficiently addresses the requests that have been made.

Senior Editor:

Comments to the Author:

Thank you for submitting your Topical Review for consideration in this Special Issue of The Journal of Physiology. We apologize for the longer than ideal time that lapsed since you submitted the manuscript--the holiday season is a particularly difficult time to secure peer Referees. The Referee whose feedback we secured believed that you've revised an important and interesting topic that is relevant to the field of research. However, the Referee and our editorial team believe that major revisions would be necessary for the manuscript to be further considered for publication. These include (but are not limited to) 1. reworking the introduction to appropriately set up the purpose of the review paper, 2. revisiting several sections that currently lack focus and provide only surface level information, 3. revising table information, and 4. overall improving the flow of the review. We would like to invite you to respond to and address each Referee critique with a revised manuscript. We look forward to seeing your revised review if you choose to submit one. Thank you again for your interest in The Journal of Physiology.

We thank the Senior Editor for her comments. We have addressed the four points made by the Senior Editor in the current version of the manuscript that we hope will satisfy her requests.

REFEREE COMMENTS

Referee #2:

In this review paper, Giacco and colleagues review the central benefits of varying ketogenic states and discuss potential mediators behind these effects, such as BDNF and PPAR delta. This review covers an important and interesting topic that is relevant to the field of research. Although the authors present good data, several major revisions are required before this review is ready for publication. Overall, this review paper was difficult to follow along to, lacks focus and is missing adequate flow.

We thank the referee for his/her acknowledgement on the topic's relevance and have restructured the review following the referee's suggestions, hoping that the changes made on both focus and flow are satisfactory.

Major points

1) The information provided in the introduction does not appropriately set up the purpose of the review paper.

- Surface level information is provided as to why Ketones are of interest

We acknowledge that the information on ketones results to be scarce and have revised the text rendering it more detailed (see page 3, line 2 -page 4 line 5).

- Further information on BDNF is required, expand on the connection between BDNF and Ketones and more so why we care about increasing BDNF levels

We addressed the relationship between BDNF, its increased expression in the ketogenic state and its beneficial effect on neurons in detail (see page 4, last 4 lines and page 5).

- Exercise and fasting are major components in this review paper yet little to no information is provided in the introduction

We thank the reviewer for this comment and have included these physiological approaches, explaining their impact on the ketogenic state (see page 3, lines 7-14).

- No information on PPAR delta is provided in the introduction

We agree with the reviewer that this information is lacking and we have now inserted information on this receptor in the introduction (See page 4, lines 6-21).

2) "Physiological interventions modulating the ketogenic state" section lacks focus

- The authors spend more time discussing organs that produce ketones rather than the actual physiological interventions that modulate a ketogenic state. • The authors should make a figure for the different organs that produce ketones and use this section to discuss how exercise, fasting and supplementation leads to increased ketone production

We acknowledge the referee's observation and have rearranged the text in order to address organs and interventions simultaneously. We changed the main header to: "Organ-specific ketone production and its modulation in response to various interventions". In accordance with the referee,

we have included a figure on the organs that produce ketone bodies (the current figure 1), and integrated the physiological interventions and ketone production in this section.

3) "Central responses to ketones, lactate or both" section provides surface level information and is scattered.

- The headings are difficult to follow along with, the authors jump from point to point with a lack of flow

We have made efforts to improve the flow of this section according to the referee's observations. The main heading now reads as follows: "Central effects of ketogenic state induction by ketogenic diets, fasting, and exercise in different organs". Subheaders are the following: "Re-shaping of the microbiome in response to ketones: beneficial central outcomes", "Skeletal muscle: a contributor in the mediation of ketone-induced centrally beneficial effects?", and "Emerging beneficial effects of brain ketone production in response to fasting"

- This is the first time the authors bring up lactate, what is the relationship between BHB and lactate? Despite lactate being an interesting topic, it does not fit within the scope of the authors review topic and should be removed.

In agreement with the referee, we only briefly mentioned this metabolite that is only produced in a particular condition, namely intense exercise, which may not be the intervention that is applied in the context of beneficially impacting neurodegeneration in a clinical context. Thus, we agree that it is out of scope and have inserted a phrase on this in the introduction (see page 5, lines 16-20).

- Why do the authors only discuss a ketogenic diet in rodents? what about humans?

We have inserted information of ketogenic diet interventions in humans (see page 16, last line - page 17, first three lines)

4) Table 1 is difficult to read and not of publication quality, authors should find a better way to present this data.

We have revised the table by better aligning the information and increasing readability avoiding the breaking of words. We also added more details on the interventions, and omitted lactate.

5) The review, after table 1, becomes heavily about PPAR delta, a lot of the information provided in this section should have been brought up in the introduction of the review. The depth of information in this section makes the reviews appear to be primarily on PPAR delta rather than the central benefits of ketogenic states.

In accordance with the referee, we have integrated a substantial part of the information on PPAR delta in the introduction, and in the subsequent chapters, in order to shift the focus of the review to the central effects that are also highlighted in the introduction, to which the sections of the physiological interventions refer in the current version. Information on agonists has been omitted.

6) The addition of the FGF21 discussion at the end of the review seems unnecessary and lacks relevance to the rest of the review. The authors are urged to narrow the scope of the review to what is directly relevant to the topic.

We have taken note of the referee's clear request to streamline the review, we removed the reference to FGF21 and focus on intervention-induced beneficial effects, leaving the bridging role of PPAR α between increased FFA supply, their oxidation, ketogenesis and transport into the brain in the flow of the review. We hope this version will be satisfactory to the referee.

Dear Dr de Lange,

Re: JP-TR-2025-287462R1 "The Central Benefit of Physiologically Induced Ketogenic States" by Antonia Giacco, Giuseppe Petito, Rosalba Senese, Maria Moreno, Assunta Lombardi, Antonia Lanni, and Pieter de Lange

Thank you for submitting your manuscript to The Journal of Physiology. It has been assessed by a Reviewing Editor and by 1 expert referee(s) and we are pleased to tell you that it is potentially acceptable for publication following satisfactory major revision.

Please address all the points raised and incorporate all requested revisions or explain in your Response to Referees why a change has not been made. We hope you will find the comments helpful and that you will be able to return your revised manuscript within 9 months. If your article is for a Special Issue, please note that we require your revised version within 2 months (rather than 9 months) in order to keep the Special issue on track. If you require longer than this, please contact journal staff: jp@physoc.org. Please note that this letter does not constitute a guarantee for acceptance of your revised manuscript.

ABSTRACT FIGURES: Authors are expected to use The Journal's premium BioRender account to create/redraw their Abstract Figures. Information on how to access this account is here:

<https://physoc.onlinelibrary.wiley.com/journal/14697793/biorender-access>.

REVISION CHECKLIST:

IMPORTANT POINTS TO NOTE WHEN REVISING YOUR MANUSCRIPT:

We look forward to receiving your revised submission.
If you have any queries, please reply to this email and we will be pleased to advise.

Yours sincerely,

Karyn Hamilton
Senior Editor
The Journal of Physiology

REQUIRED ITEMS

- Your MS must include a complete "Additional information section" with the following 4 headings and content:

Competing Interests: A statement regarding competing interests. If there are no competing interests, a statement to this effect must be included. All authors should disclose any conflict of interest in accordance with journal policy.

Author contributions: Each author should take responsibility for a particular section of the study and have contributed to writing the paper. Acquisition of funding, administrative support or the collection of data alone does not justify authorship; these contributions to the study should be listed in the Acknowledgements. Additional information such as 'X and Y have contributed equally to this work' may be added as a footnote on the title page.

It must be stated that all authors approved the final version of the manuscript and that all persons designated as authors qualify for authorship, and all those who qualify for authorship are listed.

Funding: Authors must indicate all sources of funding, including grant numbers. If authors have not received funding, this must be stated.

It is the responsibility of authors funded by RCUK to adhere to their policy regarding funding sources and underlying research material. The policy requires funding information to be included within the acknowledgement section of a paper. Guidance on how to acknowledge funding information is provided by the Research Information Network. The policy also requires all research papers, if applicable, to include a statement on how any underlying research materials, such as data, samples or models, can be accessed. However, the policy does not require that the data must be made open. If there are considered to be good or compelling reasons to protect access to the data, for example commercial confidentiality or legitimate sensitivities around data derived from potentially identifiable human participants, these should be included in the statement.

Acknowledgements: Acknowledgements should be the minimum consistent with courtesy. The wording of acknowledgements of scientific assistance or advice must have been seen and approved by the persons concerned. This section should not include details of funding.

EDITOR COMMENTS

Reviewing Editor:

Comments to the Author:

The reviewer's report indicates that although the manuscript has seen improvements, further effort is needed in terms of writing, especially concerning the flow and presentation of essential concepts. Certain ideas in the introduction and discussion require more detail, particularly the segment on skeletal muscle's role in the central benefits of ketones. While the authors discuss the relationship between exercise and muscle ketone, they fail to connect muscle ketone production with central outcomes.

Senior Editor:

Comments to the Author:

Thank you for submitting your revised Topical Review for continued consideration in this Special Issue of The Journal of Physiology. As noted previously, the Referee and Reviewing Editor agree that the topic has the potential to be impactful and they also agree that your revisions addressed some of their initial concerns. However, there are still some challenges with the flow and focus of ideas reviewed. They've provided some guidance to continue to make major revisions and we invite you to do so if you would like the Topical Review to continue to be considered for publication. Thank you for your efforts and

your interest in this Special Issue. We look forward to seeing your revised review if you choose to submit one.

REFEREE COMMENTS

Referee #2:

The Authors adequately addressed some of my initial concerns, specifically there are improvements to the introduction and table 1. Although some of the manuscript has been re-structured, there is still a major issue with the flow and focus on the manuscript.

Many of the sections only provide surface level information on the topics. For example, Skeletal muscle as a contributor to ketone-induced central benefits section; although the authors discuss the link between exercise and muscle ketone, they fail to link muscle ketone production to central effects. The authors go on to discuss exercise-induced central benefits, however this is not the purpose of this section.

Moreover, there does not seem to be any flow with the sections that are brought up. For example, the PPAR section lacks any transition from the section prior, thus making it come across as random. Overall, the authors need to better organize their manuscript.

END OF COMMENTS

REPLIES TO REVIEWING EDITOR, SENIOR EDITOR AND REFEREE'S COMMENTS

Reviewing Editor:

Comments to the Author:

The reviewer's report indicates that although the manuscript has seen improvements, further effort is needed in terms of writing, especially concerning the flow and presentation of essential concepts. Certain ideas in the introduction and discussion require more detail, particularly the segment on skeletal muscle's role in the central benefits of ketones.

- We thank the reviewing editor for his/her helpful suggestions and we have done our best to address the comments adequately. We rearranged the introduction to obtain a better flow, introducing the rationale behind the metabolic switch from glucose to fat/ketones (page 3 lines 7 and 8). In the first lines of page 4, we introduced why ketone metabolism in exercised skeletal muscle, increasing muscle integrity, is crucial for maintenance of cognition, which we develop further in the section on muscle. We also highlighted the outcomes of ketogenic diets versus other ketogenic interventions regarding FFA release (see, page 4, last part). Pages 5 and 6 introduce the factors released by muscle in response to the physiological ketogenic interventions, and underlines the interplay between organs in response to ketogenic states, tissue cross-talk related to health being the subject of the special issue.

While the authors discuss the relationship between exercise and muscle ketone, they fail to connect muscle ketone production with central outcomes.

-We realize that this section lacked clarity. In physiologically induced ketogenic states (by aerobic exercise or fasting-related interventions), the main producer of ketones is the liver. Muscle receives ketone bodies from the liver, ketones are not produced by muscle (see page 10, an initial phrase is inserted for clarity). An overview of ketone-producing organs is given in figure 1, which inserted upon request by referee #2. Ketones taken up in muscle trigger the secretion of factors that improve cognition, that may differ based on the way the ketogenic state is induced (exercise versus fasting-related interventions) and include, as discussed, BDNF, irisin, cathepsin, and thyroid hormone, which all have beneficial effects on cognition, and interact (irisin and cathepsin B both induce central BDNF expression in ketogenic states).

Senior Editor:

Comments to the Author:

Thank you for submitting your revised Topical Review for continued consideration in this Special Issue of The Journal of Physiology. As noted previously, the Referee and Reviewing Editor agree that the topic has the potential to be impactful and they also agree that your revisions addressed some of their initial concerns.

-We thank the senior editor for her encouraging comments.

However, there are still some challenges with the flow and focus of ideas reviewed. They've provided some guidance to continue to make major revisions and we invite you to do so if you would like the Topical Review to continue to be considered for publication. Thank you for your efforts and your interest in this Special Issue. We look forward to seeing your revised review if you choose to submit one.

-We thank the senior editor for having given us the opportunity to re-revise the manuscript and we hope to have addressed the comments of the reviewing editor and referee#2 to their satisfaction.

REFEREE COMMENTS

Referee #2:

The Authors adequately addressed some of my initial concerns, specifically there are improvements to the introduction and table 1.

-We are pleased the referee has seen improvements in the introduction and the table. We rearranged the introduction further creating a better flow and clarity in the order of events, in line with the new flow of the manuscript .

Although some of the manuscript has been re-structured, there is still a major issue with the flow and focus on the manuscript.

Many of the sections only provide surface level information on the topics. For example, Skeletal muscle as a contributor to ketone-induced central benefits section; although the authors discuss the link between exercise and muscle ketone, they fail to link muscle ketone production to central effects.

-We would like to emphasize that muscle does not produce ketones, except in a pathological condition namely poorly controlled diabetes, as shown in 1 publication, which is not the subject of this review (DOI: 10.1016/0026-0495(90)90163-7). We have now better explained that ketones that enter the muscle from the liver trigger induction of BDNF and other factors that are excreted from the muscle tissue and enter in the bloodstream. BDNF, key to improved cognition, is thus increased in both muscle and brain by direct action of ketones. Since BDNF is able to cross the blood brain barrier muscle-derived BDNF in response to induction of ketogenesis by the liver may be beneficial to the brain. We amended figure 2 indicating that skeletal muscle and the intestine are at the receiving end of hepatic ketone production, which subsequently leads to the production of factors that increase cognition, in addition to the direct effect of the ketones themselves. This is in line with the topic of the special issue, which concerns tissue-crosstalk. We now inserted this figure at the end of the text.

The authors go on to discuss exercise-induced central benefits, however this is not the purpose of this section.

-We apologize to the referee that we have not been sufficiently clear on this subject. We fully realize this and better explained that ketones taken up in the muscle in response to ketogenic states such as fasting and aerobic exercise trigger the secretion of factors that improve cognition, including BDNF, irisin, cathepsin B, and thyroid hormone. Irisin has been shown to be induced by ketones in a cell system, and both irisin and cathepsin B upregulate BDNF at the central level. We now restructured the muscle section focusing on ketones (page 18 and 19, green sections).

Moreover, there does not seem to be any flow with the sections that are brought up. For example, the PPAR section lacks any transition from the section prior, thus making it come across as random. Overall, the authors need to better organize their manuscript.

-We agree with the referee, and have now embedded the PPAR section at the beginning of the introduction and also in the text following the introduction, as ketogenesis starts with increased fatty acid supply to the liver and its oxidation followed by ketogenesis, all these processes being governed by the PPARs. In the next section we address the additional ketone producing organs, which provides an easier flow for the reader. We hope that this change is satisfactory to the referee.

Dear Associate Professor de Lange,

Re: JP-TR-2025-287462R2 "The Central Benefit of Physiologically Induced Ketogenic States" by Antonia Giacco, Giuseppe Petito, Rosalba Senese, Maria Moreno, Assunta Lombardi, Antonia Lanni, and Pieter de Lange

We are pleased to tell you that your paper has been accepted for publication in The Journal of Physiology.

Authors should note that it is too late at this point to offer corrections prior to proofing. Major corrections at proof stage, such as changes to figures, will be referred to the Editors for approval before they can be incorporated. Only minor changes, such as to style and consistency, should be made at proof stage. Changes that need to be made after proof stage will usually require a formal correction notice.

Yours sincerely,

Karyn Hamilton
Senior Editor
The Journal of Physiology

P.S. - You can help your research get the attention it deserves! Check out Wiley's free Promotion Guide for best-practice recommendations for promoting your work at www.wileyauthors.com/eoo/guide. You can learn more about Wiley Editing Services which offers professional video, design, and writing services to create shareable video abstracts, infographics, conference posters, lay summaries, and research news stories for your research at www.wileyauthors.com/eoo/promotion.

IMPORTANT NOTICE ABOUT OPEN ACCESS: To assist authors whose funding agencies mandate public access to published research findings sooner than 12 months after publication, The Journal of Physiology allows authors to pay an Open Access (OA) fee to have their papers made freely available immediately on publication.

You can check if your funder or institution has a Wiley Open Access Account here: <https://authorservices.wiley.com/author-resources/Journal-Authors/licensing-and-open-access/open-access/author-compliance-tool.html>.

EDITOR COMMENTS

Reviewing Editor:

Comments to the Author:

The reviewer report indicates that their previous comments have all been addressed.

Senior Editor:

Comments to the Author:

Thank you for submitting your revised Topical Review. We are pleased to accept it for publication in this Special Issue of The Journal of Physiology. Thank you for your interest in The Journal and Congratulations!

REFEREE COMMENTS

Referee #2:

Authors have addressed previous concerns.